# Unsupervised Anomaly Detection Applied to Φ-OTDR

**DOI:** 10.3390/s22176515

**Published:** 2022-08-29

**Authors:** Antonio Almudévar, Pascual Sevillano, Luis Vicente, Javier Preciado-Garbayo, Alfonso Ortega

**Affiliations:** 1ViVoLab, Aragón Institute for Engineering Research (I3A), University of Zaragoza, 50009 Zaragoza, Spain; 2Applied Physics Department, Aragón Institute for Engineering Research (I3A), University of Zaragoza, 50009 Zaragoza, Spain; 3Aragon Photonics Labs (APL) and Electronic Engineering and Communications Department, University of Zaragoza, 50009 Zaragoza, Spain

**Keywords:** distributed acoustic sensors, Φ-OTDR, deep learning, autoencoder, Unsupervised Anomaly Detection

## Abstract

Distributed acoustic sensors (DASs) based on direct-detection Φ-OTDR use the light–matter interaction between light pulses and optical fiber to detect mechanical events in the fiber environment. The signals received in Φ-OTDR come from the coherent interference of the portion of the fiber illuminated by the light pulse. Its high sensitivity to minute phase changes in the fiber results in a severe reduction in the signal to noise ratio in the intensity trace that demands processing techniques be able to isolate events. For this purpose, this paper proposes a method based on Unsupervised Anomaly Detection techniques which make use of concepts from the field of deep learning and allow the removal of much of the noise from the Φ-OTDR signals. The fact that this method is unsupervised means that no human-labeled data are needed for training and only event-free data are used for this purpose. Moreover, this method has been implemented and its performance has been tested with real data showing promising results.

## 1. Introduction

Distributed acoustic sensing comprises several techniques which make use of light–matter interaction phenomena to turn the fiber into a distributed sensor. Those based on Rayleigh scattering have been widely used because of the relatively high scattering coefficient compared to Raman or Brillouin [1]. In 1993, an enhanced version of an Optical Time Domain Reflectomer (OTDR) employing a highly coherent laser, named phase-Sensitive OTDR (Φ-OTDR), was demonstrated and tested as an intrusion detector [2,3]. Φ-OTDRs are based on the variation of the backscattered intensity trace between pulses propagating through the fiber. The jagged shape of this trace is generated by the random interference of the light backscattered from different points along the fiber. Strain or temperature variations in the fiber cause a localized change in the relative phase between scattering points that severely affect the interference at that point. Thus, changes in the profile of this trace between pulses reveal minute local variations of physical conditions at any point of the fiber. Due to this feature, Φ-OTDR has been widely employed as a high sensitivity distributed sensor in many fields, such as pipeline protection [4,5], live traffic monitoring [6] or third party intrusion (TPI) [7,8].

The performance of this technology in practical application fields, especially in TPI, rests on its ability to locate and identify threats without increasing the nuisance alarm rate. For that purpose, the system needs to provide the best possible SNR level in order to detect events accurately. In its most simple version, the detection stage of the Φ-OTDR is based on a direct-detection scheme that provides only the intensity backscattered trace [9]. These recovered intensity traces are always noisy and present high fluctuations because of the non-linear response of the interference with the phase changes induced by the stimulus [10]. Traditionally, the most employed methods to reduce this impairment are based on the demodulation of the phase from the backscattered trace. These techniques implement schemes such as I/Q demodulation [11,12], heterodyne detection [13,14] or self-correlation algorithms of the trace [15,16,17]. The first two categories are based on the beat of two or more intensity signals that have a controlled delay between them and require coherent detection and high performance acquisition, among other hardware requirements, which increases the complexity and the cost of the solution. In contrast, self-correlation algorithms keep the direct-detection scheme while recovering the phase from the intensity trace. However, these last algorithms are based on specific characteristics of the signal that appear only for certain types of stimuli, narrowing its practical field of use. Even though these techniques either demand costly hardware or cannot be applied for any previously unknown stimuli, they have been widely employed in phase-sensitive OTDR. This is because the direct-detection configuration has not been considered as an alternative due to the low SNR than can be achieved and its effect in event detection for real scenarios.

Recently, there has been a growing interest in the use of deep-learning techniques applied to distributed acoustic sensors. These techniques have been proven to enhance the performance of the direct-detection Φ-OTDR by increasing the SNR without requiring extra hardware on the optical setup [18]. Post-processing algorithms applied to the raw intensity trace have been reported in multiple studies to denoise the signal and increase the event detect ratio. In this work, we have experimentally analyzed the Unsupervised Anomaly Detection (UAD) paradigm for event detection in direct-detection Φ-OTDR. Throughout this work, we analyze the SNR-enhanced techniques already reported in the literature and evaluate the UAD algorithm for event detection in real scenarios.

This paper is organized as follows. Section 2 describes the properties of the backscattered signal registered in Φ-OTDR and the non-hardware related approaches that can be found in the literature to increase its effective SNR and introduces the novel approach presented in this work. Section 3 details the experimental setup and the field experiments performed to validate the algorithm. Section 4 explains the proposed algorithms and all the steps involved from the raw signal to its output. Section 5 shows the results of the experiments and the discussion. Finally, Section 6 presents our conclusions.

## 2. Unsupervised Anomaly Detection in Φ-OTDR

### 2.1. Signal in Direct-Detection *Φ*-OTDR

The signal received at the detector in phase-sensitive OTDR is continuous. At any precise time, the signal acquired corresponds to the coherent sum of the fields reflected by the scatter centers in the fiber located within the region stimulated by the pulse at that time. This sum can be expressed as:(1)$$E{\left(t\right)}_{z=0}=E_{0}e^{-2\alpha \bar{z}}e^{i\omega t}\sum_{m=1}^{M}r_{m}e^{i\phi_{m}}$$
where $E_{0}$ is the reference amplitude, α is the attenuation coefficient of the fiber, ω accounts for the frequency of the pulsed light and $\bar{z}$ is the physical location of the center of the pulse and is defined as $\bar{z}=\left(1/2\right)\left(tv_{g}-w_{p}/2\right)$, with $v_{g}$ being the group velocity in the fiber and $w_{p}$ being the pulse width. The summation term accounts for all the *M* scatter centers; with reflectivity $r_{m}\in \left[0,1\right]$ and $\phi_{m}$, the optical phase change causes uniform distribution in the interval [0,2π), which lies within the spatial region stimulated by half of the pulse width and thus is located in $z\in [\left(1/2\right)\left(tv_{g}-w_{p}\right),\left(1/2\right)\left(tv_{g}\right)]$ [19]. In the most general case, fiber attenuation within the pulse width can be neglected as the spatial resolution is assumed to be small enough. The intensity registered by the detector, omitting the losses, can be expressed in terms of the resulting field as
(2)$$\begin{array}{c} I={\left|E\right|}^{2}={\left|\sum_{m=1}^{M}r_{m}e^{i\phi_{m}}\right|}^{2}=\sum_{m=1}^{M}r_{m}^{2}+2\sum_{k=1}^{M-1}\sum_{l=k+1}^{M}r_{k}r_{l}\cos\left(\phi_{k}-\phi_{l}\right) \end{array}$$

The term regarding $\cos(\phi_{k}-\phi_{l})$ accounts for the relative phase change between scatter centers and can be considered non-variant for static mechanical conditions of the fiber. Phase noise of the laser source or environmental fluctuations such as temperature can affect this value, generating a slow drift than can be neglected in a short time range compared to pulse frequency. Now, if we consider a perturbation that involves a phase change $\theta_{p}$ between the reflector q−1 and *q* in a region of *M* scattering centers, the difference of the intensity signals between before, $I_{M}$, and after the perturbation, $I_{M}^{\prime }$, can be described as
(3)$$\begin{array}{c} I_{M}-I_{M}^{\prime }=2\sum_{k=1}^{q-1}\sum_{l=q}^{M}r_{k}r_{l}\left[\cos\left(\phi_{k}-\phi_{l}\right)\left(1-\cos\theta_{p}\right)-\sin\left(\phi_{k}-\phi_{l}\right)\sin\theta_{p}\right] \end{array}$$

This dependence shows the non-linear response of the phenomena to perturbations of the fiber [20]. Direct-detection schemes are based on the analysis of this intensity signal and thus this non-linearity is a strong impairment in its use for sensing due to the high nuisance alarm rate it implies when using event-detection algorithms in real scenarios [20]. The most common and simple approach to overcome this nonlinearity in the intensity trace is the implementation of a coherent stage that could reveal the phase of the backscattered signal. For that, the laser source is split and a small portion of the original signal, aided by an acousto-optic modulator, is frequency shifted and directly coupled into the detector with the signal coming from the fiber [21]. By doing this, the phase of the backscattered signal can be measured via the electric signal originating from the beat of the two optical signals, which presents a low frequency that can be handled by standard electronics. I/Q demodulation or Hilbert transform demodulation are common techniques that can be performed to recover the phase in this architecture [21,22]. Other classical approaches have been proved valid in order to recover the phase such as those based on the Kramers–Kroning receiver [23], Rayleigh backscattering self-interference detection [16] and the phase-generated carrier [24]. However, all these solutions require extra hardware or rely on some specific conditions of the signal that limit its range of application or increase its deployment cost in real scenarios.

### 2.2. Event Detection for *Φ*-OTDR

Recently, we have seen a different approach in order to increase the SNR and reduce the false alarm rate in phase-sensitive OTDR based on the application of advanced algorithms to the raw data. The use of machine-learning techniques and tools has been studied as an alternative to hardware-oriented solutions in order to enhance the performance of phase-sensitive OTDR without increasing the complexity and cost of the setup. So far, different methods based on machine-learning techniques have been proposed to process signals from DAS sensors.

Depending on the data to be worked with and the objective, two working scenarios are possible. The first aims to classify the different types of situations that the DAS can capture. That is, the objective in this scenario is to indicate whether there is an event or not and, if there is, to say what type of event it is. This scenario requires a very large amount of data of each type of event labeled by a human, which is costly and time-consuming. The second scenario, on the other hand, aims to discern between event-absence and event-presence situations. That is, in cases where events are present, there is no indication of which possible event it is. The main advantage of this scenario is that only event-free data are needed for its design and, therefore, it does not require human-provided labeling. In addition, in the case of wanting to classify events, as is the aim in the first scenario, the fact of previously separating between absence and presence of events can facilitate the task considerably. Equivalently, the objective of the latter scenario can be seen as that of denoising the signals, so that the SNR in event areas is much higher after processing them and, therefore, it is easier to detect events. This work belongs to the second scenario, but it is convenient to analyze works belonging to both groups, since some techniques used are similar.

As for the first scenario, in [25], principal component analysis (PCA) is used to obtain features of reduced dimension that are subsequently classified using support vector machine (SVM). In [26,27], convolutional neural networks (CNNs) whose input are the Mel-frequency cepstral coefficients (MFCCs) [28] of the signal corresponding to each DAS distance are used to classify different types of events. Moreover, it is proposed in [29] to use a network based on VGG16 [30] to classify events; it is trained with signals generated by Generative Adversarial Networks (GANs) [31]. Finally, in [7], CNN and the long short-term memory network (LSTM) are combined to both detect and classify events by taking spatio-temporal patches as inputs.

Regarding the second scenario, a solution based on real-time event-detection methods, in particular YOLOv3 [32], is proposed in [33]. On the other hand, in [34,35] it is proposed to generate artificial data and noise, sum them and process the result through a CNN that is able to eliminate the added noise. This idea is based on the concept of a denoising autoencoder [36]. Finally, in [37], the previous idea is combined with a GAN, the generator of which is a denoiser that takes as input the summed signal and noise. The reason for introducing a GAN to the previous proposal is that the generator is improved and overcomes the discriminator.

This work belongs to the second scenario, i.e., it tries to detect events, but not to classify them. However, the way of approaching the task is different from those described above. For this purpose, it is assumed that the events that the Φ-OTDR aims to capture are anomalies and Unsupervised Anomaly Detection (UAD) techniques are introduced. UAD refers to the task of detecting out-of-normal events using only anomaly-free data (also known as normal data) for its design. UAD techniques proposed in this work also use neural networks, as some of the previously described systems do, but the workflow is different, as is explained in Section 4.

## 3. Experimental Setup and Field Experiment

The setup employed is based on a conventional direct-detection ϕ-OTDR and it is depicted in Figure 1. The emission source is a coherent laser centered in 1550.12 nm. Due to the availability for the field test, the linewidth of the laser employed was 1 Khz, although for the direct-detection scheme this is not a requirement. The continuous wave emission is pulsed by an acousto-optic modulator fed with a square signal provided by an electrical Signal Generator (SG). The pulse width employed is 200 ns and the pulse frequency is 1 kH, which corresponds to a 20 m spatial resolution and an effective bandwidth of 500 Hz. The pulsed light is amplified by an Erbium Doped Fiber Amplifier (EDFA) and the undesired noise generated by spontaneous emission is filtered out with the Optical Bandpass Filter (OBPF) of 100 GHz conveniently centered at the emitting wavelength. The amplified and filtered pulse is coupled into the fiber under test (FUT) by means of a circulator. Finally, the backscattered signal is guided into the detection stage, where it is amplified and filtered with the same scheme as in the emission stage. At the end of the detection stage, the optical signal is registered with a photodetector (PD) and sampled by an analog–digital converter (ADC) at 50 MS/s, which yields to a sampling resolution of 2 m. The maximum reach of the system will be determined by the degradation of the OSNR along the measurement setup, and will be settled by these configuration parameters and the distributed attenuation in the layout. Due to the aforementioned nonlinearity nature of the signals in the phase-sensitive OTDR, the OSNR analysis requires a statistical analysis of the signal at the ending point where a reference stimulus is placed [38]. For this setup and the configured parameters, the maximum reach was set to 41 km.

To train and evaluate the systems explained in the next section, the layout was designed so the same stimulus could be recorded at several distances at the same time and the effect of the distance for the perturbation could be precisely evaluated. The FUT in this case was a standard single mode fiber deployed inside and optical fiber cable, which was buried in the ground and hosted several optical fibers. By placing reels of fiber with different lengths in one side of the cable, and splicing the fibers together in the other end of the cable accordingly, we could define an optical path where the same pulse travelled the event area, and thus the same physical location, several times in the same trip. In this scenario, depicted in Figure 1, we used reels of 5, 10 and 20 km, so we have the same event replicated in distances ≈ 0, 5, 15 and 35 km at the same time.

Based on the most common threats for TPI in underground infrastructures, four different events, listed in Table 1, were performed in the sensing area: *Hydraulic*— this consisted of a hydraulic hammer hitting the ground above the cable. *Digging*—this was performed with an excavator scraping in the ground and located at three different surface distances perpendicular to the fiber cable. *Compactor*—this was carried out by a plate compactor compacting the ground, and again at three different distances perpendicular to the fiber. *Moving along*—this event was recorded when the excavator was being positioned at the different locations and was passing over the cable in the surface. Although this last event is not considered an actual threat, it usually precedes others and its detection plays a key role in early warning systems. In addition, 10 min of event-free data were captured in each set to train the network. Having data captured in this controlled environment has two main advantages. The first one is that we can be sure that the data used to train Unsupervised Anomaly Detection systems are indeed free of anomalies. The second one is that the abnormal data are labeled and the time and position at which they occurred is known, so evaluation is straightforward.

## 4. Methodology

This section explains the proposed method to detect events in Φ-OTDR signals. This method uses deep-learning techniques applied to anomaly detection.

### 4.1. Deep Learning and Unsupervised Anomaly Detection

As already explained in Section 2, in the field of anomaly detection, methods based on deep-learning techniques have been proposed in recent years. In this particular case, the term “unsupervised” refers to the fact that no labels are used to train the neural networks. In particular, autoencoders [39] have been proposed for this task [40].

An autoencoder is a neural network that has two main parts—an encoder and a decoder—and whose objective is to make the input to the encoder and the output of the decoder as similar as possible by passing through a compressed version of the data, which is called code. Equivalently, this means choosing a cost function that compares input and output of the network. In particular, when working with bidimensional data such as images, it is common to choose the mean square error, i.e., given an image $X=\left(x_{ij}\right)\in R^{N\times M}$ and an autoencoder $A_{\theta }:R^{N\times M}\to R^{N\times M}$ whose parameters are θ, the loss function is given by:(4)$$L\left(X,A_{\theta }\right)=\sum_{i=1}^{N}\sum_{j=1}^{M}{\left(x_{ij}-A_{\theta }{\left(x\right)}_{ij}\right)}^{2}$$

These structures have been proposed as a method for Unsupervised Anomaly Detection [41]. For this purpose, the network is trained only with normal data in order to be able to accurately reconstruct this type of data. Thus, in the prediction stage, when the network tries to process another type of data (anomalies), the performance will be clearly worse.

As an illustrative example, an autoencoder was designed and trained to encode and decode images with a handwritten number two from the MNIST dataset [42]. Subsequently, in the prediction stage, two images serve as inputs. The first one contains a handwritten number two (Figure 2a) and, as can be seen in Figure 2b, the autoencoder reconstructs it precisely. The second image contains a handwritten number eight (Figure 2c), which the autoencoder is not able to reconstruct it properly, as can be seen in Figure 2d. Thus, we could easily detect that the eight is an anomaly by comparing the input and the output of the network.

### 4.2. *Φ*-OTDR Signal to Patches

Signals from Φ-OTDR contain time and distance information. Therefore, as a result of this, a large matrix $S=\left(s_{ij}\right)\in R^{L_{t}\times L_{d}}$ is obtained, where $s_{ij}$ contains information about instant *i* at position *j*, and $L_{t}$ and $L_{d}$ are the number of temporal and spatial samples, respectively.

The first step is to normalize the matrix so that the network training converges. To do this, $S^{N}$ is defined as:(5)$$S^{N}=\left(s_{ij}^{N}\right)\in R^{L_{t}\times L_{d}},\;s_{ij}^{N}=\frac{s_{ij}-\mu_{j}}{\sigma_{j}}$$
where $\mu_{j}$ and $\sigma_{j}$ are the mean and standard deviation at distance $j\in \{1,2,\cdots ,L_{t}$}, respectively. These values are taken from training data and they are used both for training and the prediction stage.

The input to the network is not the large matrix $S^{N}$; smaller overlapping patches are included. Figure 3 clarifies the way in which information is passed as input to the network. Thus, four parameters are defined: $N_{t}$ and $N_{d}$, which are the patch length in the temporal and spatial dimensions, respectively, and $M_{t}$ and $M_{d}$, which are the patch offset in the temporal and spatial dimensions, respectively. It follows that:(6)$$\begin{array}{c} S^{P}=\left(s_{i_{t},i_{d},j_{t},j_{d}}^{P}\right)\in R^{W_{t}\times W_{d}\times N_{t}\times N_{d}},s_{i_{t},i_{d},j_{t},j_{d}}^{P}=s_{M_{t}\left(i_{t}-1\right)+j_{t},M_{d}\left(i_{d}-1\right)+j_{d}}^{N} \end{array}$$
where $W_{t}=\left\lceil \frac{L_{t}-N_{t}}{M_{t}}+1\right\rceil$ and $W_{d}=\left\lceil \frac{L_{d}-N_{d}}{M_{d}}+1\right\rceil$.

Subsequently, $S^{P}$ is flattened into a 3D array $S^{F}$, so that each $s_{i}^{F}$ makes up the input to the network.
(7)$$\begin{array}{c} S^{F}=\left(s_{i,j_{t},j_{d}}^{F}\right)\in R^{W_{t}\cdot W_{d}\times N_{t}\times N_{d}},s_{\left(i_{t}-1\right)W_{d}+i_{d},j_{t},j_{d}}^{F}=s_{i_{t},i_{d},j_{t},j_{d}}^{P} \end{array}$$

### 4.3. Autoencoder Scheme Proposed

As explained in Section 4.1, autoencoders were used for Unsupervised Anomaly Detection. For this working scenario, the inputs to the network are the patches of size $N_{t}\times N_{d}$ obtained according to the process described previously. These inputs are processed through an autoencoder with convolutional layers to take full advantage of the spatio-temporal correlation within each patch. Only anomaly-free data are used to train the network; mean square error was chosen as the cost function and Adam with a learning rate equal to 0.0001 was used as an optimizer.

Figure 4 shows the encoder and decoder structures. The output of the encoder is used as input for the decoder. Furthermore, as can be seen, in addition to pooling and upsampling layers, we use residual blocks. The structure of these blocks is shown in Figure 5 and their purpose within the architecture is explained below.

The residual blocks scheme can be seen in Figure 5; they are based on residual networks presented in [43]. In them, the inputs to the convolutional layers are connected (as a sum generally) with their outputs. It is argued that in this way, the backpropagation algorithm does not suffer from the vanishing-gradient problem [44] and, therefore, the depth of the networks can be increased. Most current architectures introduce residual paths in some way between the inputs and outputs of their layers, improving performance on almost any task. In particular, the architecture used for this work is shown in Figure 5.

### 4.4. Reconstruction—Calculation of Anomaly Scores

Once the network has been trained, the data from which we want to obtain information concerning whether they are anomalous are processed through the network in the prediction stage. When the network output is available, the instantaneous square error between each pixel of input and output is defined as the anomaly score. This means that no resolution is lost and it is possible to have information about whether each element of the input is anomalous. Therefore, if we assume that $A_{\theta }$ is the autoencoder whose parameters are θ and $s_{i}^{F}\in R^{N_{t}\times N_{d}}$ is the input to the network, the output of the network is $A_{\theta }\left(s_{i}^{F}\right)\in R^{N_{t}\times N_{d}}$.

We can then define the anomaly marker for each patch $s_{i}^{F}$ as the instantaneous square error between it and its output from the network, i.e:(8)$$\begin{array}{c} E^{F}=\left(e_{i,j_{t},j_{d}}^{F}\right)\in R^{W_{t}\cdot W_{d}\times N_{t}\times N_{d}},\\ e_{i,j_{t},j_{d}}^{F}=|s_{i,j_{t},j_{d}}^{F}-A_{\theta }{\left(s_{i}^{F}\right)}_{j_{t},j_{d}}| \end{array}$$

Carrying out the inverse procedure to the one explained in Section 4.2, we obtain the following:(9)$$\begin{array}{c} E^{P}=\left(e_{i_{t},i_{d},j_{t},j_{d}}^{P}\right)\in R^{W_{t}\times W_{d}\times N_{t}\times N_{d}},\\ e_{i_{t},i_{d},j_{t},j_{d}}^{P}=e_{\left(i_{t}-1\right)W_{d}+i_{d},j_{t},j_{d}}^{F} \end{array}$$
(10)$$\begin{array}{c} E^{N}=\left(e_{ij}^{N}\right)\in R^{L_{t}\times L_{d}},\\ e_{M_{t}\left(i_{t}-1\right)+j_{t},M_{d}\left(i_{d}-1\right)+j_{d}}^{N}=e_{i_{t},i_{d},j_{t},j_{d}}^{F} \end{array}$$
(11)$$\begin{array}{c} E=\left(e_{ij}\right)\in R^{L_{t}\times L_{d}},\\ e_{ij}=\sigma_{j}\cdot e_{ij}^{N} \end{array}$$

Therefore, the anomaly score *E* is obtained, which is a matrix of the same size as the signal coming from the DAS *S* and which provides information on whether or not an event has occurred at each of the points of the signal. Thus, when $e_{ij}$ is greater than a threshold, an event is considered to have occurred at instant *i* and distance *j*.

## 5. Results

This section presents the results obtained for the database we worked with. As explained in Section 2, the resolution of this problem can be seen from two perspectives. The first one consists of treating the task as an event-detection problem, while the second one consists of treating it as a denoising problem. Therefore, when it comes to showing and analyzing the results, a metric reflecting the performance for each of the two perspectives will be presented. For the first one, the area under the ROC curve (AUC) [45] will be used. AUC can be interpreted as the probability that a binary classifier system correctly identifies two samples from different classes. For this particular case, it indicates the probability that, given normal and anomalous data, the system is able to tell which is which. For the second one, the improvement in terms of the SNR (ΔSNR) will be used. ΔSNR is the difference between SNR_E_ and SNR_S_, which are the SNR for the prediction error and the SNR for the signal coming from the sensor, respectively. To calculate SNR_E_ and SNR_S_, the average power of the event zones is divided by the average amplitude in the non-event zones *E* and *S*, respectively. Due to the fact that we have different types of events to measure performance, each of these scores is going to be specified for each type of event. In addition, since the signal power decreases with distance from the DAS, the detection capability and SNR can be expected to degrade with distance.

To carry out these two types of analyses, some design parameters have been chosen empirically. In concrete terms, it has been chosen that $N_{t}=256$, $M_{t}=256$, $N_{d}=8$, $M_{d}=6$. The AUC and ΔSNR for different distance sections are presented in Table 2 and Table 3, respectively.

Two conclusions can be drawn from Table 2. The first one is that some stimuli are more easily detectable than others. For example, *Hydraulic* or *Digging* are more easily detectable than *Compactor* or *Moving along*. It is also easier to detect events that have occurred closer to the DAS than those that have occurred further away. In particular, a considerable decrease is observed in the 20 to 35 km range compared to the others. Table 3 shows conclusions similar to the two previous ones, but referring to the SNR improvement. In particular, here it is especially remarkable that a much lower SNR increase is achieved in the 20 to 35 km section compared to the others. The fact that this improvement in SNR is obtained is due to the fact that the system reconstructs the event-free zones (noise) much more accurately than the event zones (signal), since it has been trained to reconstruct the event-free zones. Therefore, the signal prediction error is much higher in the event zones than in the event-free zones.

If we analyze each of the events, we can draw different conclusions that are of interest. The first is that the *Hydraulic* event is almost always detectable even at the highest distances. On the other hand, in the *Moving along* event there is a clear degradation of the detection capability for distances greater than 20 km. For the events produced by *Digging*, quite similar performances are obtained for those produced at 0, 5 and 10 m from the fiber. Finally, for the events produced by *Compactor*, there is a significant degradation of the system for the 10-meter distance compared to the 0- and 5-meter distances.

In addition, Figure 6 and Figure 7 show a comparison between the input and output of the proposed anomaly-detection system for the *Hydraulic* event for distances of 5 and 35 km. It can be seen in these figures that the event is more easily detectable at 5 km than at 35 km from the DAS. In these figures it can be seen that the events (yellow zone) are easily detectable after applying our processing, while before the processing the signal in the event zone is even weaker than in the non-event zone. Furthermore, it can be seen that the magnitude of the prediction error is larger in the 5 km event than in the 35 km event, which is due to the direct dependence between attenuation and distance. These figures complement Table 2 and Table 3. First, it can be perfectly perceived how the detection capacity of the events is very high, since the event and event-free zones are distinguished with a great contrast. On the other hand, it can be seen how the power level is practically similar in the event and noise zones in the original signal, so that the SNR is around 0 dB, while in the denoised signal the power difference is much higher.

Finally, Figure 8a shows the histogram of the normal data and the anomalous data corresponding to each trench before the proposed processing. It can be easily observed that only some of the events close to the sensor can be detected and that all those occurring at distances greater than 5 km are completely indistinguishable with respect to the normal data. Therefore, without signal processing, the intensity trace has no use in detecting events at distances beyond 5 km. On the other hand, Figure 8b shows the histogram of the data after processing. Here, it can be seen that only some data for events occurring at 35 km from the sensor are confused with the normal situation and that the overlap between events at distances less than 35 km and the normal situation is null. Comparing these two histograms, it is possible to understand the importance of introducing the proposed processing, since it allows to considerably increase the range of distances with which the sensor is able to work.

## 6. Conclusions

This paper has presented a method to detect events in signals obtained from direct-detection DAS Φ-OTDR sensors. The method is based on considering the potential events as anomalies and using anomaly-detection techniques to detect them. For this purpose, unsupervised deep-learning systems using only event-free signals were used. The main advantage of this approach compared to other techniques is that no human-provided labels are needed. This loosening of the requirements for the training set widens the usability of this technique for event detection in real environments without previous knowledge of the potential events.

To train the neural networks, event-free data corresponding to 10 min and 35.2 km of fiber were used. On the other hand, to test the performance of the proposed method, a small database with different intentionally triggered and controlled events was also used. This method provides an effective noise reduction from the signals coming from the Φ-OTDR that enables precise detection of the aggression events up to 35 km. All the results obtained in this work were analyzed in terms of the AUC and the SNR improvement that applying the proposed method provides.

## Figures and Tables

**Figure 1 sensors-22-06515-f001:**
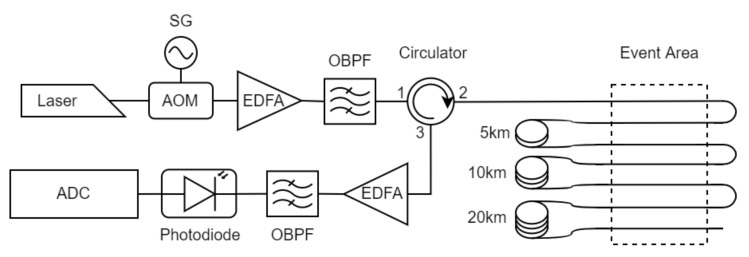
Setup.

**Figure 2 sensors-22-06515-f002:**
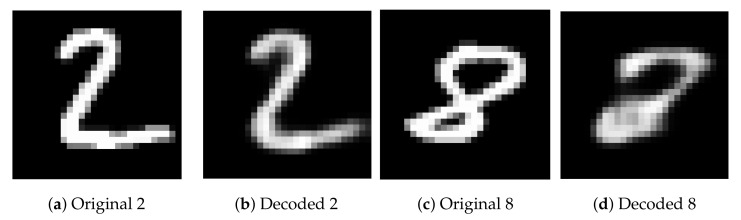
Comparison of reconstruction of a handwritten number two and a handwritten number eight with an autoencoder trained to reconstruct images with a handwritten number two.

**Figure 3 sensors-22-06515-f003:**
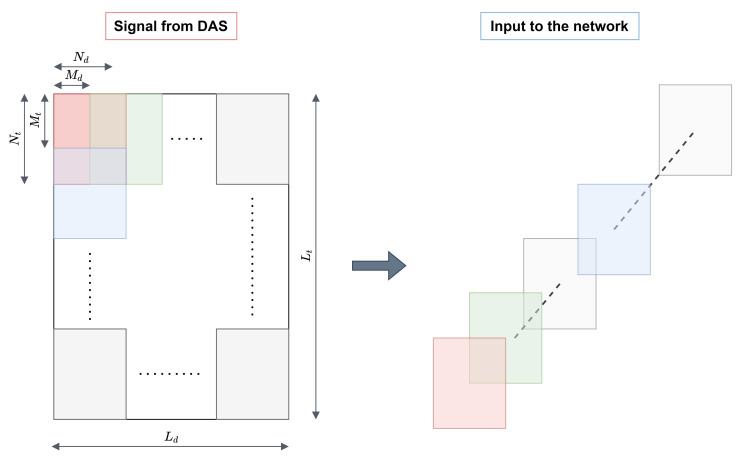
The signal coming from the Φ-OTDR does not serve as input directly to the network, but fixed size patches are taken in.

**Figure 4 sensors-22-06515-f004:**
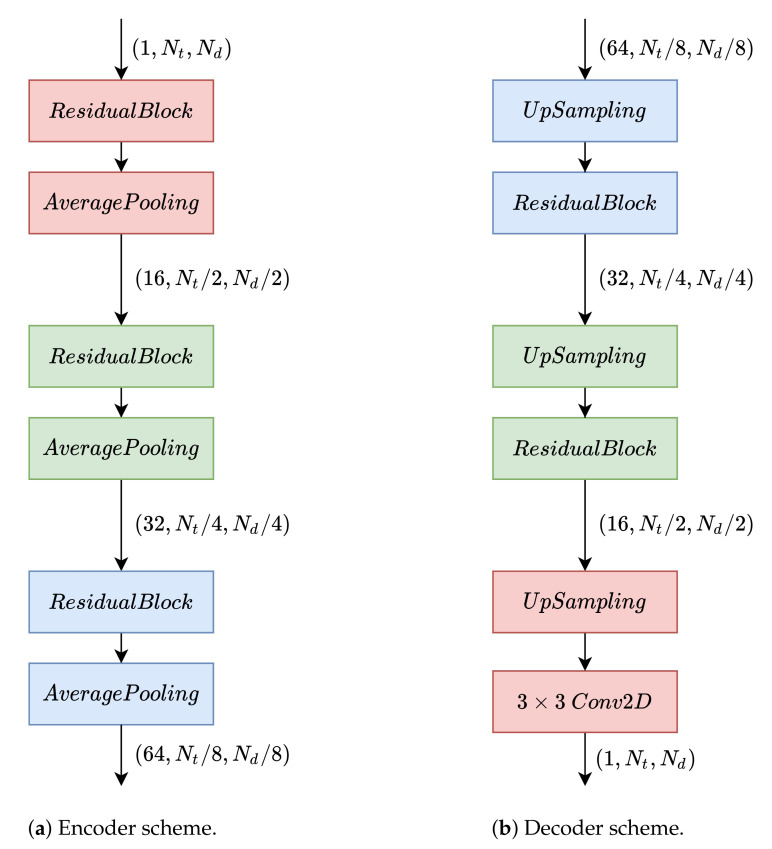
Autoencoder architecture used to obtain anomaly scores.

**Figure 5 sensors-22-06515-f005:**
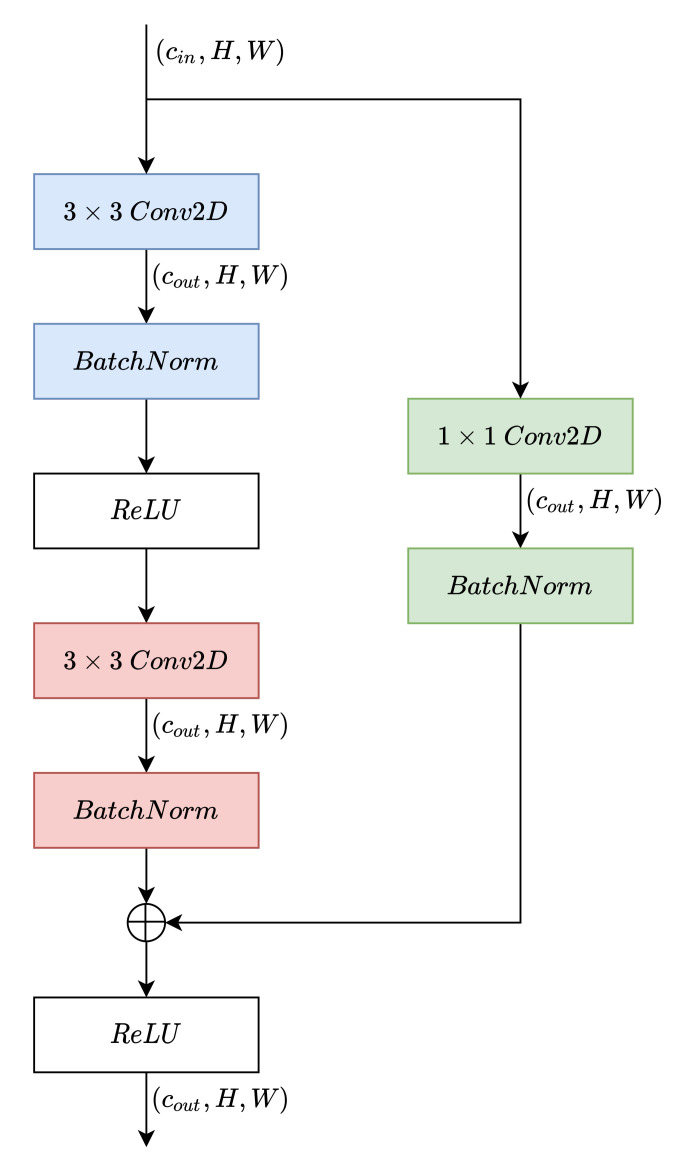
Residual blocks used to cause the autoencoder to conform.

**Figure 6 sensors-22-06515-f006:**
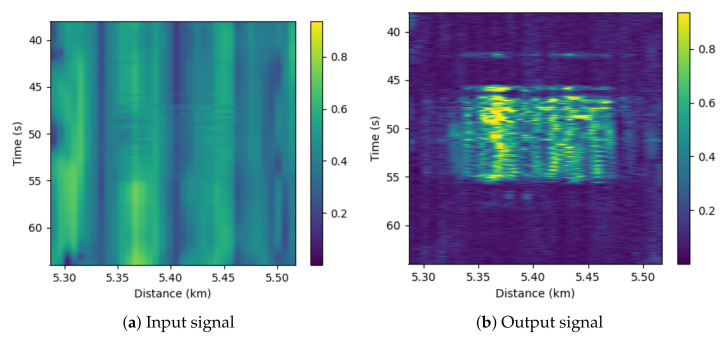
Comparison of the original and denoised signals around 5 km ditch for *Hydraulic*.

**Figure 7 sensors-22-06515-f007:**
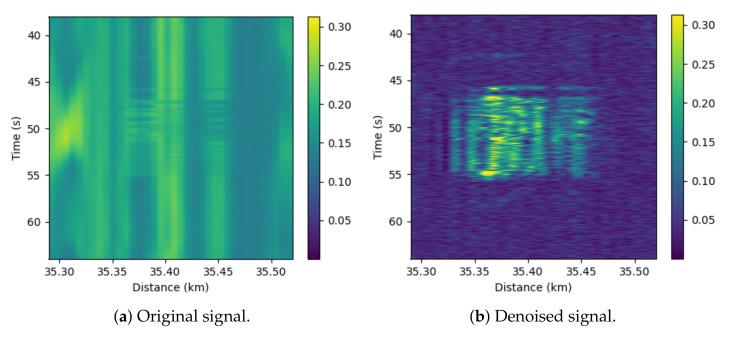
Comparison of the original and denoised signals around 35 km ditch for *Hydraulic*.

**Figure 8 sensors-22-06515-f008:**
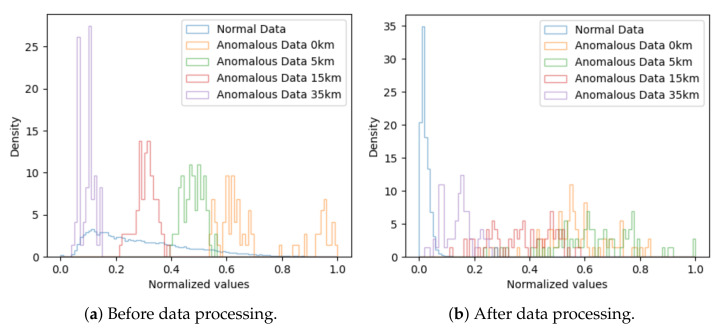
Comparison of histograms before and after processing the data through our system for *Hydraulic*.

**Table 1 sensors-22-06515-t001:** Events performed that form the test dataset.

Event ID	Description of the Event
Hydraulic	Hydraulic hammer working
Moving along	Heavy machinery moving along the ground
Digging 0 m	Excavator digging right on top of fiber
Digging 5 m	Excavator digging from 5 m of fiber
Digging 10 m	Excavator digging from 10 m of fiber
Compactor 0 m	Removed soil is added back and compacted right on top of the fiber
Compactor 5 m	Removed soil is added back and compacted 5 m from the fiber
Compactor 10 m	Removed soil is added back and compacted 10 m from the fiber

**Table 2 sensors-22-06515-t002:** AUC (%) for different events and distances.

Event ID	0–2.5 km	2.5–10 km	10–20 km	20–35.2 km
Hydraulic	99.994	99.997	99.997	99.954
Moving along	99.343	99.523	99.677	91.167
Digging 0 m	99.997	99.988	99.934	95.301
Digging 5 m	99.924	99.959	99.981	95.606
Digging 10 m	99.932	99.947	99.971	98.631
Compactor 0 m	97.587	98.820	98.474	74.853
Compactor 5 m	97.127	97.968	98.556	78.776
Compactor 10 m	94.849	98.191	95.505	52.326

**Table 3 sensors-22-06515-t003:** ΔSNR (dB) for different events and distances.

Event ID	0–2.5 km	2.5–10 km	10–20 km	20–35.2 km
Hydraulic	13.80	15.00	13.40	9.77
Moving along	11.58	10.49	10.02	5.63
Digging 0 m	13.10	12.78	11.84	4.53
Digging 5 m	13.63	12.51	12.27	3.72
Digging 10 m	14.58	13.22	12.00	3.97
Compactor 0 m	7.64	7.42	6.09	1.36
Compactor 5 m	11.03	9.21	9.05	3.75
Compactor 10 m	5.92	4.50	4.21	0.489

## Data Availability

The data used for the development of this system have been obtained within the framework of the project. At the moment, they are not available online. For more information on the data, please contact the authors.

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
