# Peer review of "Unsupervised Anomaly Detection Applied to Φ-OTDR"

_sensors, 2022, doi:10.3390/s22176515_

Round 1
Reviewer 1 Report
The manuscript concerns the detection of events in signals obtained from direct detection DAS Φ-OTDR sensors; deep Learning techniques are applied to anomaly detection where no labels are used to train a neural networks: The technique includes auto-encoders as a method for unsupervised anomaly detection.
The network is trained only with normal data in order to be able to accurately reconstruct this type of data. In the prediction stage, when the network tries to process another type of data (anomalies), the performance will be worse.
Two different approaches are used for the anomalies detection
1) the area under the ROC curve (AUC) is used as the probability that a binary classifier system correctly identifies two samples from different classes; it indicates the probability that, given a normal and an anomalous data, the system is able to tell which is which
2) The improvement in terms of the SNR (ΔSNR) is used. ΔSNR is the difference between SNRE and SNRS, the SNR for the prediction error and the SNR for the signal coming from the sensor, respectively.
In my opinion the paper is well written, and the presented technique is an useful tool for the evaluation of the sensor performance.
I suggest the following revisions:
1) The authors should include more details on the DAS sensor, such as Sampling resolution,Spatial Resolution, Maximum Reach, Sampling rate
2) Different events and distances are reported for the measurements and different approaches.
The authors should discuss deeply the results in the different events, in terms of different performance and results
Reviewer 2 Report
· This paper proposes a method based on UAD to remove much of the noise from the Φ-OTDR signals. It is interesting that no human-labeled data are needed for training to improve the signal noise ratio (SNR). Presentation and writing is good. There are several materials should be supplemented before it can be accepted.
· 1. In section 4, the authors explained the proposed method to detect events by using Deep Learning techniques applied to anomaly detection. However, the analysis of SNR improvement is lack, which should be added to support the experimental data in Table.3, Figure. 6 and 7.
· 2. In the section 5, experimental results verify the effectiveness of the method to a certain extent. However, the improvement of the SNR in Table.3 lacks an explanation and the supporting materials. Further, the authors need to point out the preconditions for the validity and shortcomings in the discussion.
